# The Isolation of Anaerobic and Facultative Anaerobic Sulfate-Reducing Bacteria (SRB) and a Comparison of Related Enzymes in Their Sulfate Reduction Pathways

**DOI:** 10.3390/microorganisms11082019

**Published:** 2023-08-06

**Authors:** Jing Wang, Xiaohong Li, Fang Guan, Zhibo Yang, Xiaofan Zhai, Yimeng Zhang, Xuexi Tang, Jizhou Duan, Hui Xiao

**Affiliations:** 1College of Marine Life Sciences, Ocean University of China, Qingdao 266003, China; 2Key Laboratory of Marine Environmental Corrosion and Biofouling, Institute of Oceanology, Chinese Academy of Sciences, No. 7 Nanhai Road, Qingdao 266071, China; 3Laoshan Laboratory, Qingdao 266000, China

**Keywords:** inner rust layer and sediment, sulfate-reducing bacteria, anaerobic SRB, facultative anaerobic bacteria, sulfate reduction pathway

## Abstract

Sulfate-reducing bacteria (SRB) are an important group of microorganisms that cause microbial corrosion. In this study, culturable SRB were isolated and identified from the inner rust layer of three kinds of steel and from sediments, and a comparison of amino acid sequences encoding related enzymes in the sulfate reduction pathway between anaerobic and facultative anaerobic SRB strains was carried out. The main results are as follows. (1) Seventy-seven strains were isolated, belonging to five genera and seven species, with the majority being *Desulfovibrio marinisediminis*. For the first time, *Holodesulfovibrio spirochaetisodalis* and *Acinetobacter bereziniae* were separated from the inner rust layer of metal, and sulfate reduction by *A. bereziniae*, *Virgibacillus dokdonensis*, and *Virgibacillus chiguensis*, etc., was also demonstrated for the first time. (2) Three strains of strictly anaerobic bacteria and four strains of facultative anaerobic bacteria were screened from seven bacterial strains. (3) Most of the anaerobic SRB only contained enzymes for the dissimilatory sulfate reduction pathway, while those of facultative anaerobic bacteria capable of producing hydrogen sulfide included two possible ways: containing the related enzymes from the dissimilatory pathway only, or containing enzymes from both dissimilatory and assimilation pathways. This study newly discovered that some bacterial genera exhibit sulfate reduction ability and found that there are differences in the distribution of enzymes related to the sulfate reduction pathway between anaerobic and facultative anaerobic SRB type trains, providing a basis for the development and utilization of sulfate-reducing bacterial resources and furthering our understanding of the metabolic mechanisms of SRB.

## 1. Introduction

SRB are a group of microorganisms that utilize inorganic sulfate as an external electron acceptor in the oxidation of energy substrates, resulting in the production of hydrogen sulfide via dissimilatory sulfate reduction [1], which is very important in the biogeochemical cycle of the sulfur and microbial desulfurization process [2]. SRB are widely distributed in marine sediments and in rust layers on metal surfaces [3] and are the most influential and dominant corrosion-accelerating agents driving metal microbial corrosion (MIC) in marine environments [4]. The hydrogen sulfide and sulfide produced during the growth of SRB directly promote metal corrosion [5]. SRB also have high hydrogenase activity, thereby producing hydrogen via formate fermentation, and they have substantial potential in the field of energy generation [6]. In addition, SRB play key roles in the biological sulfur cycle; they can be used to precipitate heavy metal ions from wastewater via the sulfur compounds produced through microbial desulfurization [7] and have been widely used for bioremediating wastewaters containing heavy metals [7] and acid mine drainage [8].

There are many kinds of SRB. More than 60 genera and more than 220 species of SRB have been reported thus far [9], including *Desulfovibrio*, *Desulfomonas*, *Desulfobulbus*, and *Desulfotomaculum*. Traditionally, SRB are considered strictly anaerobic bacteria [10], but several studies have confirmed that some bacteria can survive in aerobic environments and show oxygen tolerance [11,12,13]. These bacteria mainly belong to the genus *Desulfovibrio*, such as *Desulfovibrio vulgaris* [14,15]. In addition, some representatives of other genera, including *Kurthia Gibsonii*, *Pseudomonas aeruginosa* [16], and *Citrobacter amalonaticus* TB10 [17], had also been considered facultative anaerobic SRB; however, they are not included in the internationally accepted and more scientific classification system of SRB [18]. Compared with anaerobic SRB, facultative anaerobic SRB grow and reproduce more rapidly and are more robust and easier to apply in industrial production settings, such as during biological desulfurization and wastewater treatment [18]. However, at present, studies on the diversity of SRB have mainly focused on the diversity of SRB taxa in different habitats [19,20,21] and the effects of environmental factors on the abundance and communities of SRB [22], while studies on the understanding of the groups and corrosion mechanisms of facultative anaerobic SRB are insufficient [23].

Microbial sulfate reduction processes mainly include assimilatory sulfate reduction and dissimilatory sulfate reduction [24], as shown in Figure 1. The sulfate assimilation pathway is widely present in living organisms and is a common reaction pathway in aerobic organisms [25,26], while the dissimilatory sulfate pathway is a unique pathway in SRB [3,27]. The process shared by these two pathways is sulfate ion activation by ATP sulfurase to APS; there are different enzymes in the assimilation pathway and the dissimilation pathway, which can be reflected by the different key enzymes involved in these two pathways. These key enzymes include (1) APS kinase (APSK), 3′-phosphoadenosine 5′-phosphosulfate reductase (PAPS reductase), and assimilatory sulfate reductase (ASiR) in the assimilatory sulfate reduction pathway and (2) adenosine-5′-phosphosulfate reductase (APS reductase) and dissimilatory sulfite reductase (Dsr) in the dissimilatory sulfate reduction pathway. Information on enzymes related to the sulfate reduction pathway can be found on the KEGG website (http://www.kegg.jp/, accessed on 26 April 2023), and the amino acid sequences encoding the target enzymes, i.e., the genetic codes for these enzymes, can be found in the proteomes of the downloaded standard type strains. However, previous studies have focused separately on the Dsr of the dissimilatory sulfate reduction pathway [28], as well as the enzymes in the assimilation pathway, such as sulfite reductase (NADPH) [29], but have rarely reported the similarities and differences in the distribution between the related enzymes in the sulfate reduction pathway of anaerobic SRB and facultative anaerobic SRB.

In this study, the composition of culturable SRB in the inner rust layer of the different steel materials and sediments was compared, and anaerobic and facultative anaerobic SRB were screened. Additionally, the distribution pattern of sulfate-reduction-related enzymes in the multiple type strains of anaerobic and facultative anaerobic SRB obtained from the different sources was analyzed. These results further our understanding of the distribution and types of SRB on metal materials and provide some clues for further understanding the metabolic mechanisms of SRB, especially those of facultative anaerobic SRB.

## 2. Materials and Methods

### 2.1. Sample Collection

In November 2016, samples were collected from steel materials that were completely immersed in the coastal seawaters of Sanya (18°17′58″ N, 109°15′18″ E), Hainan (Figure 2). The inner rust layers of a carbon steel (Q235) [30], low-alloy steel (Q345), and low-carbon high-strength alloy steel (921A) specimen were sampled with a sterile scraper to scrape off the rust layers. The layers were approximately 1–2 mm thick and looked like soft mud. After removing the dark gray middle rust layer, the small amount of black rust found closest to the steel surface was sampled with a sterile scraper to scrape the rust layers, which were quickly transferred into 10 mL sterile plastic centrifuge tubes. Each sample in the centrifuge tubes was a mixture of three independent samples collected from an approximately 5 × 5 cm area of the same steel plate [31]. At the same time, samples from the sediments of the seabed below these alloy plates were collected [32]. All samples were stored in a foam box filled with dry ice and transported to the laboratory within 12 h for immediate processing [33]. Hereafter, “Q235,” “Q345,” “921A”, and “sediment” refer to the names of the corresponding samples.

### 2.2. Preparation of Bacterial Suspension of the Samples from the Rust Layer and Sediment

Subsequently, 2 g of each inner rust layer sample and sediment sample was accurately weighed and added to a triangular flask containing 18 mL of sterile seawater, 7 glass beads, and Tween-80 (to a final volume ratio of 1:200,000 *v*/*v*), and a suspension was prepared by shaking the mixture in a shaking incubator (HZQ-C, Harbin Dongming Medical Instrument Factory, Harbin, China) at 100 rpm for 30 min. The bacterial suspension was then prepared into a 10^−1^, 10^−2^, and 10^−3^ series of dilutions.

### 2.3. Separation, Purification, and Cryopreservation of Anaerobic SRB

In a portable glove box filled with N_2_ (Captairpyr, Erlab Company, Val-de-Reuil, France), which ensured that no oxygen was present in the operating environment, the prepared bacterial suspension was spread on Postgate’s B (PGB) modified medium (anaerobic medium for SRB) plates [30,34] containing KH_2_PO_4_ (0.5 g·L^−1^), NH_4_Cl (1.0 g·L^−1^), Na_2_SO_4_ (1.0 g·L^−1^), CaCl_2_·2H_2_O (0.1 g·L^−1^), MgSO_4_·7H_2_O (2.0 g·L^−1^), C_3_H_5_O_3_Na (80%) (3.5 mL·L^−1^), yeast extract (1.0 g·L^−1^), FeSO_4_·7H_2_O (0.5 g·L^−1^), agar (20.0 g·L^−1^), vitamin C (0.1 g·L^−1^), and H-Cys-OH·HCl (0.5 g·L^−1^) with a pH between 7.0 and 7.2. These plates were sealed and put into 2.5 L round-bottomed vertical anaerobic culture bags with AnaeroPack (Mitsubishi Gas Chemical Co., Inc., Tokyo, Japan) and then cultured at 25 °C for 72 h. When the medium turned black, this indicated the presence of SRB. Then, plates of a suitable dilution were selected, and the colonies grown on them were isolated and purified by streaking several times to obtain single colonies that could darken the medium [35]. Those strains that were able to blacken the medium and had a rotten egg smell were considered to be SRB [36] and were cultured in anaerobic Postgate’s B (PGB) modified liquid medium in penicillin bottles [30] to carry out subsequent research.

### 2.4. Preliminary Identification of Strains

Genomic DNA was extracted using an Ezup column genomic DNA extraction kit [30]. The 16S rDNA was amplified using bacterial universal primers 27F and 1492R, and the samples were transferred to Sangon Biotech (Shanghai) Co., Ltd. (Shanghai, China), for 16S rDNA gene sequencing. The National Center for Biotechnology Information (NCBI) website (https://www.ncbi.nlm.nih.gov/, accessed on 26 April 2023) was referenced for sequence analysis to obtain the standard strains most similar to the tested strains. Neighbor-joining phylogenetic trees were constructed using the bootstrap method in MEGA 6.06, and the number of bootstrap replications was 1000.

### 2.5. Screening of Facultative Anaerobic SRB

The bacteria were inoculated on modified PGB medium plates in a sterile environment and cultured at 25 °C in a biochemical incubator (SPX-150, Wanfeng Instrument Co., Ltd., Changzhou, China) for 72 h. Due to their cultivation under aerobic conditions, SRB strains that could grow on the plate under these conditions can be considered facultative anaerobic SRB, while bacteria that could not grow on the plate medium are considered anaerobic SRB.

### 2.6. Distribution of Related Enzymes in the Sulfate Reduction Pathway

In total, 23 sulfate-reducing bacterial type strains were analyzed, including 7 strains of SRB obtained from this study, 9 strains of SRB preserved in our laboratory [30] (3 anaerobic; 6 facultative anaerobic), and 7 strains of SRB (3 anaerobic; 4 facultative anaerobic) reported in the literature (Table 1). Based on the 16S rDNA results obtained from the identification of the strain, the name of the strain types with the highest degree of similarity and highest coverage for the 23 SRB strains were determined using Ezbiocloud 16S database (ChunLab, Inc., Seoul, Republic of Korea) (https://www.ezbiocloud.net/identify, accessed on 26 April 2023).

Information on the enzymes related to the sulfate reduction pathway can be found on the KEGG website (http://www.kegg.jp/, accessed on 26 April 2023). The genomes and proteomes of the 23 strain types were downloaded from the NCBI website and used to perform the isozyme clustering of sulfate-reduction-related genes. The amino acid sequences encoding the target enzyme were searched in the proteome of the downloaded strains and the target enzymes. BioEdit Sequence Alignment Editor software (version 7.0.9.0) and the NCBI website (https://www.ncbi.nlm.nih.gov/, accessed on 26 April 2023) were used to perform the alignment of amino acid sequences offline and online, respectively, and the amino acid sequences were edited using EditSeq software (Lasergene 7, DNASTAR Inc., Madison, WI, USA). The amino acid sequences of target enzymes encoded by different bacteria were compared using CLUSTALX 1.83 software and MEGA 6.06 software, respectively. The methods mentioned above were similar to those referenced in a previous study [30]. The results for the distribution of related genes in the sulfate reduction pathway obtained from the sulfate-reducing bacterial type strains were used to indicate the distribution pattern in our test strains.

## 3. Results

### 3.1. Identification of Isolated Bacteria

In total, 137 strains of bacteria were isolated in this study, of which 96 strains blackened the culture medium, and 77 strains were successfully activated and identified as belonging to seven species, five genera, and three phyla (Table 2). The dominant phylum and genus were *Proteobacteria* and *Desulfovibrio*, respectively, and *D. marinisediminis* dominated at the species level, accounting for approximately 61% of the total number of bacteria isolated.

The results showed that 60 strains of bacteria were obtained from the different inner metal rust layers, which were distributed over three phyla, five genera, and seven species (Table 2), while 17 strains were obtained from the sediments, distributed over two phyla, four genera, and four species. The dominant species of bacteria in the sediments were the same as those in the inner rust layers. Most of the isolated bacterial species were obtained from Q345.

### 3.2. Screening of Facultative Anaerobic SRB

Among the seven isolates of culturable SRB, three were strictly anaerobic, namely *D. marinisediminis*, *H. spirochaetisodali*, and *V. dokdonensis*, and four were facultative anaerobic bacteria, namely *V. chiguensis*, *A. bereziniae*, *P. bellariivorans*, and *P. denitrificans*. Through this separation, it was found that the number of anaerobic SRB in the inner rust layers and sediments was significantly greater than that of facultative anaerobic SRB (*p* < 0.05, ANOVA test). *D. marinisediminis* and *H. spirochaetisodalis*, which were isolated from both the rust layers and sediments, were anaerobic bacteria and could not grow in aerobic environments. There were a few facultative anaerobic strains in each sample, but the species were different. Among them, Q345 had the greater number of anaerobic sulfate-reducing bacterial species, followed by 921A (Table 3). The facultative anaerobic SRB *V. chiguensis* was isolated only from Q235, while *A. bereziniae* could be obtained from other locations. The facultative anaerobic SRB *P. bellariivorans* and *P. denitrificans* were only isolated from Q345. *D. marinisediminis* was isolated from Q235, 921A, Q345, and the sediment samples.

Notably, during the process of screening for facultative anaerobic SRB, although the facultative anaerobic SRB survived in an aerobic environment and grew colonies on the plates, unlike in an anaerobic culture, they did not blacken the medium, i.e., no black precipitates of FeS were produced (Figure 3).

The isolated and screened bacteria were numbered. The facultative anaerobic SRB were labeled with an “F”, the anaerobic SRB were labeled with an “A”; “V”, “A”, “P”, “D”, and “H” referred to the different genera, and these letters were placed in front of the numbers to distinguish between them. The phylogenetic tree for the seven bacterial strains is shown in Figure 4. It was found that the anaerobic SRB and facultative anaerobic SRB were mostly clustered separately, and their phylogenetic differences were noticeable in their evolutionary relationships.

### 3.3. Isozyme Clustering Results

Isozyme clustering of the sequences of genes related to the sulfate reduction pathway in the strains of 23 SRB (nine strains of strict anaerobic bacteria and 14 facultative anaerobic bacteria) showed that the genes encoding ATP sulfatase in both pathways could be found in the sequences of the 23 strains of SRB (Figure 5). Surprisingly, we found genes encoding reductases for the dissimilatory sulfate reduction pathway in all strictly anaerobic and facultative anaerobic SRB strains tested, but these typical sulfate-reducing strains, usually classified as SRB, had genes encoding both APS reductase and Dsr, whereas other strains, including one strictly anaerobic strain, *V. dokdonensis*, and 12 facultative anaerobic strains, such as *V. chiguensis*, *A. bereziniae*, *P. bellariivorans*, and *P. denitrificans*, had genes encoding Dsr only, but no genes encoding APS reductase. In addition, among the facultative anaerobic bacteria, *P. bellariivorans*, *P. denitrificans*, *A. bereziniae*, *D. desulfuricans*, and *D. oxyclinae* only had genes for Dsr and no genes for enzymes related the assimilatory sulfate reduction pathway, while two of the strictly anaerobic strains, *V. dokdonensis* and *D. nigrificans*, had genes encoding both Dsr and assimilation-pathway-related enzymes.

## 4. Discussion

### 4.1. Composition of Culturable SRB in Rust Layers and Sediments

In this study, 77 strains of SRB were isolated from inner rust layers and sediments, all of which were previously reported, with none being new species. However, *H. spirochaetisodalis* and *A. bereziniae* were found in inner metal rust layers for the first time, and the sulfate-reducing and facultative anaerobic properties of *A. bereziniae*, *V. dokdonensis*, *V. chiguensis*, *P. bellariivorans*, and *P. denitrificans* were identified for the first time. *H. spirochaetisodalis* and *A. bereziniae* were found in the inner rust layer of a metal for the first time.

The culturable SRB isolated from the inner rust layers and marine sediments were essentially the same, and the dominant phyla were Proteobacteria and Firmicutes, which was consistent with the results of the high-throughput sequencing of Q235 samples and sediments from the same location [32]. Bacteria belonging to Proteobacteria are often closely associated with corrosion and are commonly found in such environments [43]. Firmicutes have also been observed to be present in large quantities in rust layer biofilms in previous reports; prior research has also shown that many strains in this phylum are associated with corrosion [32,44]. In addition, a small population of Bacteroidetes was isolated in this study. Li [35] also isolated this phylum from rust layers sampled in the same location, and the results of high-throughput sequencing also showed the existence of this phylum [32].

At the genus level, *Desulfovibrio* was the most abundant genus. The bacteria belonging to *Desulfovibrio* were isolated from both the inner metal rust layer and the sediments, which was consistent with the bacterial diversity results in the steel rust layer detected using high-throughput sequencing [30,32]. The bacteria of this genus are often the main cause of microbial corrosion [45]. The steel corrosion ability of *Desulfovibrio* has been widely studied in laboratory studies [46,47]. Liu and Häggblom [48] found that *D. marinisediminis* exhibits reductive dehalogenation ability, and its debromination activity is not inhibited by sulfate. Li [44] reported the existence of this bacterium in a Q235 rust layer, which was the first time that this bacterium was reported from isolation studies of internal rust layers and sediments associated with steel materials.

*Halodesulfovibrio* also accounted for a large proportion of the culturable SRB isolated from the inner rust layer and sediments. Previous studies have also reported its sulfate reduction properties. Although this genus is closely related to *Desulfovibrio*, there are few reports on the discovery of *Halodesulfovibrio* in marine rust layers. *Halodesulfovibrio* was found in heavy metal acidic wastewaters and coastal marine sediments from the eutrophic Tokyo Bay of Japan [49]. However, it was the first time that *H. spirochaetisodalis* was found in metal rust layers in our study.

The genus *Virgibacillus* was mostly isolated from saltworks in previous reports, and there are few reports of this genus in corrosive rust layers. In addition, nitrate reduction and sulfate reduction pathways have been reported in this genus. Some bacteria of this genus have also been reported to produce H_2_S [50,51]. Previous studies have shown that *Virgibacillus* spp. is a salt-tolerant species that can survive in high-permeability environments and can also be used for wastewater treatment [52]. This may be the reason why bacteria of this genus can survive in rust layer environments.

*A. bereziniae* has been reported to be a pathogen in previous studies. This species demonstrates chemical heterotrophic metabolism, which makes it possible for these bacteria to compete for nutritional resources. Previous studies have mainly focused on the multidrug resistance of this genus [53]. However, until now, there have been no reports of the discovery of *Acinetobacter* in inner rust layers, and no research has been put forth on its corrosive effects. However, a member of this genus was described as sulfate-reducing bacteria [54]. In this study, the sulfate-reducing property of this bacterium was discovered for the first time.

Only a very small proportion of the genus *Prolixibacter* was isolated in this study, and Iino [55] reported that *P. denitrificans* has nitrate reduction pathways and that *P. denitrificans* isolated from crude oil is corrosive to iron (Fe-0) [56]. However, contemporary research on this genus of bacteria is still limited; thus far, there have been few reports on the effect of the strains on corrosion, and the effect of this species on corrosion is difficult to evaluate.

The abundance of microorganisms varies on metal surfaces and sediment. The bacterial composition of the inner rust layers and sediments isolated during this study is consistent with the results of the high-throughput sequencing of bacteria in metal rust layers and sediments conducted by Zhang [32] at the same sampling location. The results show that more bacterial species are present in the rust layers of alloy samples relative to those in marine environments, which may be due to the enrichment of microorganisms in the oligotrophic marine environment by metal alloys and due to the promotion of the growth of biofilm-related microorganisms [57].

### 4.2. Screening of Facultative Anaerobic Culturable SRB

Previous studies have shown that SRB can use organic matter on metal surfaces as a carbon source in the absence of oxygen or with very little oxygen and use hydrogen produced within the bacterial biofilm to reduce sulfate to hydrogen sulfide and obtain energy for survival from the redox reaction [58]. Moreover, our study also confirmed that there are indeed some bacteria that can reduce sulfate in the absence of oxygen but can also survive in the presence of oxygen.

Both anaerobic and facultative anaerobic bacteria existed in our samples, of which three strains were strictly anaerobic bacteria, namely *D. marinisediminis*, *H. spirochaetisodalis*, and *V. dokdonensis*, and four strains were facultative anaerobic bacteria, namely *P. bellariivorans*, *P. denitrificans*, *A. bereziniae*, and *V. chiguensis*. The number of anaerobic SRB separated from the inner rust layer and sediments was significantly larger than that of facultative anaerobic bacteria, and the proportion of anaerobic SRB separated from the inner rust layer was relatively large. Ke [59] also found that facultative anaerobic bacteria reached higher concentrations during the early stages of corrosion, but anaerobic SRB gradually gained ground in the later stages. We speculate that there is less oxygen in the inner rust layer, which limits the utilization of oxygen by facultative anaerobic bacteria.

*D. marinisediminis* and *H. spirochaetisodalis* have been reported to be strictly anaerobic SRB [60], and *V. dokdonensis* has also been reported to survive in anaerobic environments [51]. In addition, *P. bellariivorans*, *P. denitrificans*, and *V. chiguensis* have all been reported to be facultative anaerobic bacteria [41,55]. All of these results are consistent with the results of this study. However, Nemec [61] isolated *A. bereziniae* from human clinical specimens and identified it to be an aerobic bacterium. For the first time, this study indicated that this bacterium can also survive in an anaerobic environment and function as a facultative anaerobic bacterium.

SRB are usually considered to be strictly anaerobic [10,62]. When studying the tolerance of SRB to oxygen, it was found that some SRB could survive in the aerobic environment, and it was considered that these SRB were facultative anaerobic bacteria [11,12,13]. However, Li [30] screened some aerobic heterotrophic bacteria that can reduce sulfate under anaerobic conditions to produce hydrogen sulfide, including *Pseudodesulfovibrio*, *Vibrio*, *Photobacterium*, and *Staphylococcus*. This study isolated bacteria from the immersed inner rust layer of steel and sediment samples in an anaerobic environment and found that some of them could survive in normal atmospheric oxygen concentrations, such as *V. chiguensis*, *A. bereziniae*, *P. bellariivorans*, and *P. denitrificans*. However, some cannot, such as *D. marinisediminis*, *H. spirochaetisodali*, and *V. dokdonensis*. This indicates that different isolation methods affect the species of SRB obtained and also indicates that bacteria with different levels of oxygen tolerance and the ability to reduce sulfate to produce hydrogen sulfide can coexist in submerged metals and sediments. However, further research is needed on the function of these bacteria in steel corrosion and their interactions.

In addition, it is necessary to point out that the four strains of facultative anaerobes can reduce sulfate under anaerobic conditions and can survive under aerobic conditions. However, these strains were not included in the previously reported, internationally accepted taxa of SRB [18]. Some facultative anaerobic strains, such as *Kurthia Gibsonii*, *Pseudomonas aeruginosa* [16], and *Citrobacter amalonaticus* TB10 [17], have been reported as facultative anaerobic SRB. However, some facultative anaerobic bacteria, such as *Shewanella* sp., can produce H_2_S under anaerobic sulfate (SO_4_^2−^), thiosulfate (S_2_O_3_^2−^), sulfite (SO_3_^2−^), four-sulfate (tetrathionate), or elemental sulfur (S) conditions. These species do not contain Dsr and do not belong to the SRB group [63]. Therefore, the bacteria that can reduce sulfur and produce hydrogen sulfide isolated from the samples in this study need to be further tested to determine whether Dsr enzymes are present.

However, according to the current methods and studies on the isolation and identification of SRB [1], most strains can be identified as SRB if the medium is blackened. In this study, although most of the strains screened and isolated from biofilms using this method were SRB of the dissimilating reduction pathway, there was also a small number of strains not widely known to produce hydrogen sulfide. These bacteria are distributed in several genera and have the same ability to produce hydrogen sulfide, although whether they have the dissimilating pathway needs to be further confirmed. However, a large number of these bacteria and *Desulfovibrio* coexisted on corrosion biofilms and belonged to the other genus. Their roles and mechanisms in metal corrosion need to be further studied.

### 4.3. Distribution of Enzymes Involved in the Sulfate Reduction Pathway of Culturable SRB

Facultative anaerobic bacteria with sulfate reduction functions have been identified in many studies, many of which have been isolated from metal corrosion samples [32], although they were not included in the reported scientific classification system of SRB. The results of the genes coding the enzymes for the sulfate reduction pathway of anaerobic SRB and facultative anaerobic SRB showed that all the strains studied had genes encoding Dsr, indicating that these strains had the dissimilatory sulfate reduction pathway.

However, most of the strictly anaerobic strains had only the enzymes required for the dissimilatory sulfate reduction pathway, except that *D. nigrificans* and *V. dokdonensis* had the enzymes required for both pathways.

The results were more complex for facultative anaerobic SRB. Three facultative anaerobic strains (including *P. bellariivorans* and *P. denitrificans*) and two scientifically classified SRB (*D. desulfuricans* and *D. oxyclinae*) only had genes coding enzymes for the dissimilatory sulfate reduction pathway and none of the genes coding the enzymes in each step of the assimilation pathway, suggesting that unless new sulfate-assimilation pathways are available, these strains may have only one sulfate reduction pathway, the dissimilatory sulfate reduction pathway. However, most facultative anaerobic bacteria, such as *V. owensii*, *C. amalonaticus*, and *E. cloacae*, had genes encoding enzymes of two pathways, which means that they can reduce sulfate via both the assimilation and dissimilation pathways; this confirmed that they had the ability to reduce sulfate through the dissimilating pathway under anaerobic conditions. These strains differed from the reported facultative anaerobic bacteria *Shewanella* sp., which does not contain Dsr [63] and is only capable of sulfate assimilation reduction and cannot be considered SRB. Therefore, these facultative anaerobic SRB in this study were able to produce hydrogen sulfide by reducing sulfate under anaerobic conditions through the dissimilation pathway and could be identified as SRB. Moreover, these strains were not reported to be scientifically classified as SRB, either anaerobic or facultative anaerobic, and did not exhibit APS reductase, suggesting that the enzymes required for the critical step in the dissimilatory sulfate reduction pathway from APS to SO_3_^2−^ may not be the same as those known and that other isozymes may exist.

In summary, we found that the taxa of SRB can be expanded and many strains with a dissimilatory sulfate reduction pathway can be classified as SRB and that the sulfate reduction mechanisms of SRB, especially of facultative anaerobic SRB, are very complex, and there may be unknown reaction pathways and reaction enzymes. However, since this study was based only on the 16S rDNA sequence of the type strains, more definitive conclusions need to be confirmed through a whole-gene sequence examination and sulfate reductase activity detection.

## 5. Conclusions

In this study, a group of SRB was isolated and identified, and the distribution of these bacteria in the inner layers of rust was compared with that in sediment samples. Three bacterial species were separated for the first time from the inner rust layer of metal, and the sulfate-reducing and facultative anaerobic properties of four bacterial species were identified for the first time. In addition, an analysis of the reductase of multiple strains of SRB showed that the facultative anaerobic SRB differed in terms of the distribution of related enzymes in the pathways of assimilation and dissimilation sulfate reduction. The sulfate reduction pathways of facultative anaerobic bacteria are more complex, and there may be unknown reaction pathways and reaction enzymes. The results of this study have furthered the understanding of SRB and provided new theories for explaining the corrosion mechanisms of strictly anaerobic and facultative anaerobic SRB.

Further research is needed to determine the production of Dsr in anaerobic and facultative anaerobic SRB under anaerobic conditions to determine whether the isolated SRB possess the dissimilation sulfate reduction pathway necessary to produce hydrogen sulfide. Moreover, differences in the mechanisms of sulfate reduction between facultative anaerobic SRB with only the dissimilation pathway and those with both pathways, as well as functional differences and interactions between anaerobic bacteria and facultative anaerobic bacteria in metal corrosion, need to be clarified.

## Figures and Tables

**Figure 1 microorganisms-11-02019-f001:**
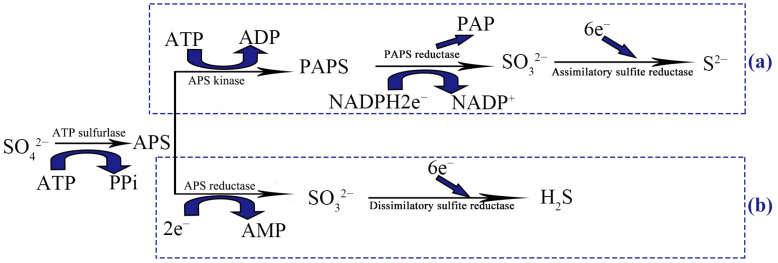
Schematic diagram of (**a**) the pathway of assimilatory sulfate reduction and (**b**) the pathway of dissimilatory sulfate reduction. The process shared by these two pathways is sulfate ion activation by ATP sulfurase to APS, for which there are different enzymes in the assimilation pathway and the dissimilation pathway, which can be reflected by the different key enzymes involved in these two pathways.

**Figure 2 microorganisms-11-02019-f002:**
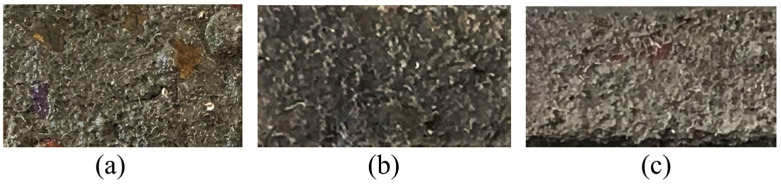
The corrosion morphology of three alloys immersed in the coastal seawaters of Sanya: (**a**) carbon steel (Q235) (10~15 mm thick); (**b**) alloy steel (Q345) (8~10 mm thick); (**c**) low-carbon quenched and tempered steel (921A) (10~15 mm thick).

**Figure 3 microorganisms-11-02019-f003:**
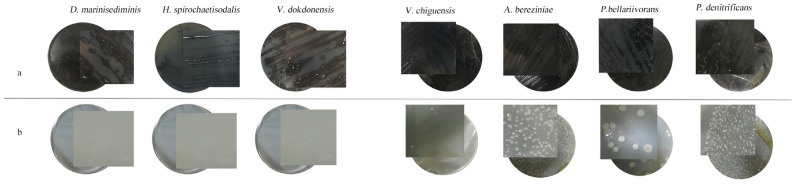
The growth of seven SRB strains inoculated on modified PGB medium plates cultured in anaerobic and aerobic environments: (**a**) anaerobic environments; (**b**) aerobic environments.

**Figure 4 microorganisms-11-02019-f004:**
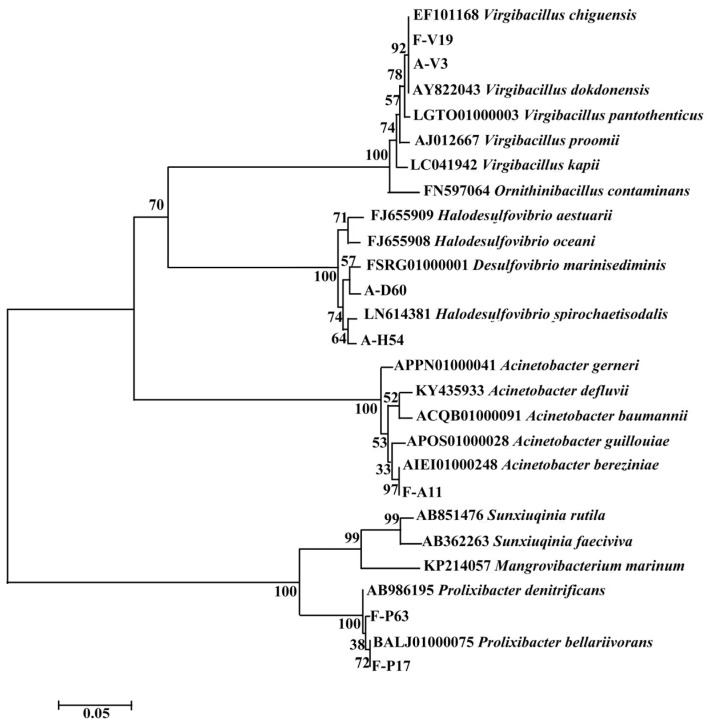
The phylogenetic tree of seven isolated SRB strains (three anaerobic SRB strains and four facultative anaerobic SRB strains) based on 16S rRNA sequences using neighbor-joining methods. Associated taxa were clustered in the bootstrap test (1000 replicates), and the bootstrap values were greater than 50%.

**Figure 5 microorganisms-11-02019-f005:**
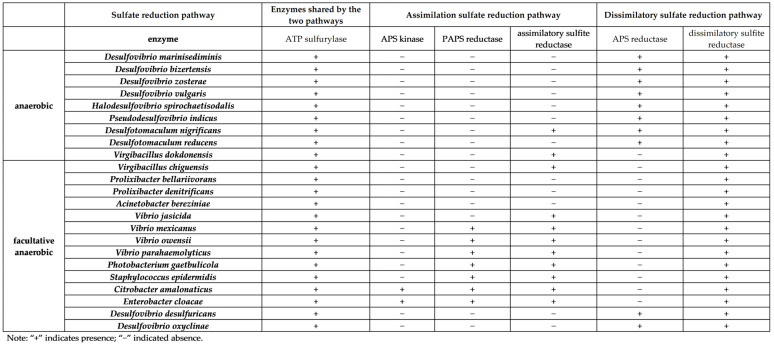
Distribution of genes encoding different sulfate reduction pathways in bacterial strains.

**Table 1 microorganisms-11-02019-t001:** Information on SRB maintained in our laboratory and SRB reviewed in the literature for the comparison of related enzymes in the sulfate reduction pathway.

Order	Characteristic	Standard Strain	Accession	Reference
1	Anaerobic	*Desulfovibrio bizertensis*	Y18049	[30]
2	Anaerobic	*Desulfovibrio zosterae*	DQ422859	[30]
3	Anaerobic	*Pseudodesulfovibrio indicus*	CP014206	[30]
4	Anaerobic	*Desulfotomaculum nigrificans*	PRJNA46699	[37]
5	Anaerobic	*Desulfovibrio vulgaris*	PRJNA395924	[38]
6	Anaerobic	*Desulfotomaculum reducens*	PRJNA13424	[39]
7	Facultative anaerobic	*Vibrio jasicida*	AB562589	[30]
8	Facultative anaerobic	*Vibrio mexicanus*	JQ434105	[30]
9	Facultative anaerobic	*Vibrio owensii*	JPRD01000038	[30]
10	Facultative anaerobic	*Vibrio parahaemolyticus*	BBQD01000032	[30]
11	Facultative anaerobic	*Photobacterium gaetbulicola*	CP005974	[30]
12	Facultative anaerobic	*Staphylococcus epidermidis*	L37605	[30]
13	Facultative anaerobic	*Citrobacter amalonaticus*	PRJNA752145	[17]
14	Facultative anaerobic	*Enterobacter cloacae*	PRJNA688591	[40]
15	Facultative anaerobic	*Desulfovibrio desulfuricans*	PRJNA666287	[41]
16	Facultative anaerobic	*Desulfovibrio oxyclinae*	PRJNA169818	[42]

**Table 2 microorganisms-11-02019-t002:** Distribution of the numbers of culturable SRB strains in the inner rust layer of different steel materials and sediments.

Phylum	Genera	Species	Q345	Q235	921A	Sediment	Total
Proteobacteria	*Desulfovibrio*	*Desulfovibrio marinisediminis*	23	5	11	8	47
	*Halodesulfovibrio*	*Halodesulfovibrio spirochaetisodalis*	1	2	9	6	18
	*Acinetobacter*	*Acinetobacter bereziniae*	1	/	1	1	3
Firmicutes	*Virgibacillus*	*Virgibacillus dokdonensis*	1	/	/	/	1
		*Virgibacillus chiguensis*	/	1	3	2	6
Bacteroidetes	*Prolixibacter*	*Prolixibacter bellariivorans*	1	/	/	/	1
		*Prolixibacter denitrificans*	1	/	/	/	1
**Sum**			28	8	24	17	77

**Table 3 microorganisms-11-02019-t003:** Distribution of the numbers of isolated bacterial strains with different respiration patterns in different materials.

Characteristic	Q345	Q235	921A	Sediment	Total	Species	Label
Facultative anaerobic	3	1	4	3	11	*V. chiguensis*	F-V19
						*A. bereziniae*	F-A11
						*P. bellariivorans*	F-P17
						*P. denitrificans*	F-P63
Anaerobic	25	7	20	14	66	*V. dokdonensis*	A-V3
						*D. marinisediminis*	A-D60
						*H. spirochaetisodalis*	A-H54

## Data Availability

The results for the current bacterial cultures are available in the NCBI SRA repository under the BioProject ID PRJNA824677: SAMN27410675.

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
