# Peer review of "The Isolation of Anaerobic and Facultative Anaerobic Sulfate-Reducing Bacteria (SRB) and a Comparison of Related Enzymes in Their Sulfate Reduction Pathways"

_microorganisms, 2023, doi:10.3390/microorganisms11082019_

Round 1

Reviewer 1 Report

The work presented in this manuscript is interesting and can be published, however, minor corrections are required as listed.

1. Abstract need to be revised focusing what is new and highlight achivemement in numbers.

2. Introduction need to be enriched with potential reports in the last 3 three years.

3. Figure 4 need to be improved.

4. Compartive approaches for isolation need to stated.

5. Discussion need depth with clarififcation of mechansictic action of isolation which schemes

English need revsison addressing the typos and mistakes

Author Response

    We greatly appreciate your questions and suggestions which you have taken precious time out of your busy schedule. These comments make us realize the flaws in the manuscript and are all valuable and very helpful for improving our manuscript and research. We have made revisions accordingly. In the following, we give a point-by-point reply to your comments and revisions were marked up using blue color in the revised manuscript.

Q1: Abstract need to be revised focusing what is new and highlight achivemement in numbers.

A1: Thank you very much for your suggestion.

       As suggested, the abstract has been reviewed and rewritten as follows (Lines 15-32):

        Sulfate-reducing bacteria (SRB) are an important group of microorganisms that cause microbial corrosion. In this study, culturable SRB were isolated and identified from the inner rust layer of three kinds of steel and from sediments, and a comparison of amino acid sequences encoding related enzymes in the sulfate reduction pathway between anaerobic and facultative anaerobic SRB strains was carried out. The main results are as follows: (1) Seventy-seven strains were isolated, belonging to five genera and seven species, with the majority being Desulfovibrio marinisediminis. For the first time, Holodesulfovibrio spirochaetisodalis and Acinetobacter bereziniae were separated from the inner rust layer of metal, and sulfate reduction by A. bereziniae, Virgibacillus dokdonensis, and Virgibacillus chiguensis, etc., was also demonstrated for the first time. (2) Three strains of strictly anaerobic bacteria and four strains of facultative anaerobic bacteria were screened from seven bacterial strains. (3) Most of the anaerobic SRB only contained enzymes for the dissimilatory sulfate reduction pathway, while those of facultative anaerobic bacteria capable of producing hydrogen sulfide included two possible ways: containing the related enzymes from the dissimilatory pathway only, or containing enzymes from both dissimilatory and assimilation pathways. This study newly discovered that some bacterial genera exhibit sulfate reduction ability, and found that there are differences in the distribution of enzymes related to the sulfate reduction pathway between anaerobic and facultative anaerobic SRB type trains, providing a basis for the development and utilization of sulfate-reducing bacterial resources and furthers our understanding of the metabolic mechanisms of SRB.

Q2: Introduction need to be enriched with potential reports in the last 3 three years.

A2: Thank you very much for your valuable suggestion.

       We have made modifications and corresponding supplements in the newly submitted version:

①…,which are very important in the biogeochemical cycle of the sulfur and microbial desulfurization process [2] (Lines 39-40);

②The hydrogen sulfide and sulfide produced during the growth of SRB directly promote metal corrosion [5] (Lines 43-44);

③However, at present, studies on the diversity of SRB have mainly focused on the diversity of SRB taxa in different habitats [19-21] and the effects of environmental factors on the abundance and communities of SRB [22], while studies on the understanding of the groups and corrosion mechanisms of facultative anaerobic SRB are insufficient [23] (Lines 62-66);

④However, previous studies have focused separately on the Dsr of the dissimilatory sulfate reduction pathway [28], as well as the enzymes in the assimilation pathway, such as sulfite reductase (NADPH) [29]. (Lines 81-83)

     In addition, we have added 14 published literatures which enriched this Introduction in lines 447-449, 454-456, 464-469, 476-478, 492-505, 510-519 (references 2, 5, 9, 10, 13, 19-23, 26-29).

Newly added References:

2. Dong, Y.; Wang, J.; Gao, Z.; Di, J.; Wang, D.; Guo, X.; Hu, Z.; Gao, X.; Wang, Y. Study on growth influencing factors and desulfurization performance of sulfate reducing bacteria based on the response surface methodology. ACS Omega 2023, 8(4), 4046-4059. https://doi.org/1021/acsomega.2c06931.

5. Hirano, S.I.; Ihara, S.; Wakai, S.; Dotsuta, Y.; Otani, K.; Kitagaki, T.; Ueno, F.; Okamoto, A. Novel Methanobacterium strain induces severe corrosion by retrieving electrons from Fe0 under a freshwater environment. Microorganisms 2022, 10(2), 270. https://doi.org/3390/microorganisms10020270.

9. Herman, J.P.; Redfern, L.; Teaf, C.; Covert, D.; Michael, P.R.; Missimer, T.M. Cumene contamination in groundwater: observed concentrations, evaluation of remediation by sulfate enhanced bioremediation (SEB), and public health issues. Int J Environ Res Public Health 2020, 17(22), 8380. https://doi.org/10.3390/ijerph17228380.

10. Tripathi, A.K.; Saxena, P.; Thakur, P.; Rauniyar, S.; Samanta, D.; Gopalakrishnan, V.; Singh, R.N.; Sani, R.K. Transcriptomics and functional analysis of copper stress response in the sulfate-reducing bacterium Desulfovibrio alaskensis Int J Mol Sci 2022, 23(3), 1396. https://doi.org/10.3390/ijms23031396.

13. Bitenieks, K.; Bārdule, A.; Eklöf, K.; Espenberg, M.; Ruņģis, D.E.; Kļaviņa, Z.; Kļaviņš, I.; Hu, H.; Lībiete, Z. The Influence of the degree of forest management on methylmercury and the composition of microbial communities in the sediments of boreal drainage ditches. Microorganisms 2022, 10(10), 1981. http://doi.org/10.3390/microorganisms10101981.

19. Ma, Q.L.; Du, H.; Liu, Y.; Li, M. Sulfate-reducing prokaryotes in mangrove wetlands: diversity and role in driving element coupling. Acta Microbiologica Sinica 2022, 62(12), 4606-4627. https://doi.org/13343/j.cnki.wsxb.20220569. (in Chinese)

20. Zou, S.B.; Gao, Q.; Cheng, H.H.; Ni, M.; Xu, Q.; Liu, M.; Zhou, D.; Zhou, Z.M.; Yuan, J.L. Vertical distribution of bacterial, sulfate-reducing and sulfur-oxidizing bacterial communities in sediment cores from freshwater prawn (Macrobrachium rosenbergii) aquaculture pond. Acta Microbiologica Sinica 2022, 62(07), 2719-2734. https://doi.org/10.13343/j.cnki.wsxb.20210690. (in Chinese)

21. Cheng, Y.; Song, T.L.; Tian, X.G.; Wang, Y.H. Distribution characteristics and environmental significance of sulfate-reducing bacterial community in high arsenic groundwater from different depth of Hetao Plain, China. Acta Microbiologica Sinica 2022, 62(06), 2372-2388. https://doi.org/10.13343/j.cnki.wsxb.20220216. (in Chinese)

22. Zhan, P.F.; Huang, J.F.; Yu, C.X.; Tong, C. Effects of saltwater and Fe (Ⅲ) pulses on community structure and abundance of methanogens and sulfate-reducing bacteria in tidal freshwater marsh of the Min River estuary. Acta Scientiae Circumstantiae 2020, 40(07), 2599-2610. https://doi.org/10.13671/j.hjkxxb.2020.0066.

23. Cui, Y.Y.; Qin, Y.X.; Ding, Q.M.; Gao, Y.N. Study on corrosion behavior of X80 steel under stripping coating by sulfate reducing bacteria. BMC Biotechnol 2021, 21(1), 5. https://doi.org/10.1186/s12896-020-00664-5.

26. Jin, Z.; Wang, W.; Li, X.; Zhou, H.; Yi, G.; Wang, Q.; Yu, F.; Xiao, X.; Liu, X. Structure and function of piezophilic hyperthermophilic Pyrococcus yayanosii Int J Mol Sci 2021, 22(13), 7159. https://doi.org/10.3390/ijms22137159.

27. Schwarz, A.; Gaete, M.; Nancucheo, I.; Villa-Gomez, D.; Aybar, M.; Sbárbaro, D. High-rate sulfate removal coupled to elemental sulfur production in mining process waters based on membrane-biofilm technology. Front Bioeng Biotechnol 2022, 10, 805712. https://doi.org/10.3389/fbioe.2022.805712.

28. Schweitzer, H.D.; Smith, H.J.; Barnhart, E.P.; McKay, L.J.; Gerlach, R.; Cunningham, A.B.; Malmstrom, R.R.; Goudeau, D.; Fields, M.W. Subsurface hydrocarbon degradation strategies in low- and high-sulfate coal seam communities identified with activity-based metagenomics. NPJ Biofilms Microbiomes 2022, 8(1), 7. https://doi.org/ 10.1038/s41522-022-00267-2.

29. Wei, Z.; Zhang, Z.; Zhao, W.; Yin, T.; Liu, X.; Zhang, H. Overexpression of MET4 leads to the upregulation of stress-related genes and enhanced sulfite tolerance in Saccharomyces uvarum. Cells 2022, 11(4), 636. https://doi.org/10.3390/cells11040636.

Q3: Figure 4 need to be improved.

A3: Thank you very much for your advice.

       We had changed the figure 4 with a clearer version. (Lines 228-232, Fig 4).

Q4: Compartive approaches for isolation need to stated.

A4: Thank you for this valuable recommendation.

   The difference between the two approaches for isolation methods was anaerobic or aerobic environments. The relevant parts have been rewritten in the revised manuscript in Lines130-141 and Lines152-157, respectively.

Separation of anaerobic SRB:

    In a portable glove box filled with N2 (Captairpyr, Erlab company, France), which ensured that no oxygen was present in the operating environment, the prepared bacterial suspension was spread on Postgate's B (PGB) modified medium (anaerobic medium for SRB) plates [30,34]. These plates were sealed and put into 2.5 L round-bottomed vertical anaerobic culture bags with AnaeroPack (Mitsubishi Gas Chemical Co., Inc., Tokyo, Japan), then stored at 25°C for 72 h. When the medium turned black, this indicated the presence of SRB. Then, plates of suitable dilution were selected and the colonies grown on them were isolated and purified by streaking several times to obtain single colonies that could darken the medium [35]. Those strains that were able to blacken the medium and had a rotten egg smell were considered to be SRB [36], and were cultured in anaerobic Postgate's B (PGB) modified liquid medium in penicillin bottles [30] to carry out the subsequent research. (Lines 130-141, Materials and Methods 2.3)

Separation of Facultative anaerobic SRB:

    The bacteria were inoculated on modified PGB medium plates in a sterile environment, and cultured at 25°C in a biochemical incubator (SPX-150, Wanfeng Instrument Co.Ltd, China) for 72 h. Due to its cultivation under aerobic conditions, SRB strains that could grow on the plate under these conditions can be considered facultative anaerobic SRB, while bacteria that could not grow on the plate medium are considered anaerobic SRB. (Lines 152-157, Materials and Methods 2.5)

Reference (Lines 520-521, 529-534):

30. Li, X.H. Screening of anaerobic and facultative anaerobic sulfate-reducing bacteria in corrosive steel layer and comparison of their corrosion properties. PhD Thesis, Ocean University of China, China. 2019. (in Chinese)

34. Postgate, J.R. The Sulphate Reducing Bacteria. 2nd edition.; Cambridge University Press, New York, 1983; pp. 224

35. Li, X.H.; Xiao, H.; Zhang, W.; Li, Y.; Tang, X.; Duan, J.; Yang, Z.; Wang, J.; Guan, F.; Ding, G. Analysis of cultivable aerobic bacterial community composition and screening for facultative sulfate-reducing bacteria in marine corrosive steel. J Oceanol Limnol 2019, 37(2), 600-614. https://doi.org/10.1007/s00343-019-7400-1

36. Pan, J.; Shao, Z.; Cao, H.; Sheng, Y. Review on the isolation and purfication methods of sulfate reducing bacteria. J Microbiol 2007, 27(5), 79-83. https://doi.org/10.3969/j.issn.1005-7021.2007.05.018. (in Chinese)

Q5: Discussion need depth with clarification of mechanistic action of isolation which schemes.

A5: Thank you very much for your insightful questions and suggestions.

        We have added the clarification of mechanistic action of isolation (Lines 341-354) which schemes in the discussion section accordingly as follows:

        SRB are usually considered to be strictly anaerobic [10,62]. When studying the tolerance of SRB to oxygen, it was found that some SRB could survive in the aerobic environment, and it was considered that these SRB were facultative anaerobic bacteria [11-13]. However, Li [30] screened some aerobic heterotrophic bacteria that can reduce sulfate under anaerobic conditions to produce hydrogen sulfide, including Pseudodesulfovibrio, Vibrio, Photobacterium, and Staphylococcus. This study isolated bacteria from the immersed inner rust layer of steel and sediment samples in an anaerobic environment, and found that some of them could survive in normal atmospheric oxygen concentrations, such as V. chiguensis, A. bereziniae, P. bellariivorans, and P. denitrificans. However, some cannot, such as D. marinisediminis, H. spirochaetisodali, and V. dokdonensis. This indicates that different isolation methods affect the species of SRB obtained, and also indicates that bacteria with different levels of oxygen tolerance and the ability to reduce sulfate to produce hydrogen sulfide can coexist in submerged metals and sediments. However, further research is needed on the function of these bacteria in steel corrosion and their interactions.

        Moreover, the new 3 relevant literatures have been added to the references (References 10, 13, 62, Lines 467-469, 476-478, 596-598).

Newly added References (References 10, 13, 62, Lines 467-469, 476-478, 596-598):

10. Tripathi, A.K.; Saxena, P.; Thakur, P.; Rauniyar, S.; Samanta, D.; Gopalakrishnan, V.; Singh, R.N.; Sani, R.K. Transcriptomics and functional analysis of copper stress response in the sulfate-reducing bacterium Desulfovibrio alaskensis Int J Mol Sci 2022, 23(3), 1396. https://doi.org/10.3390/ijms23031396.

13. Bitenieks, K.; Bārdule, A.; Eklöf, K.; Espenberg, M.; Ruņģis, D.E.; Kļaviņa, Z.; Kļaviņš, I.; Hu, H.; Lībiete, Z. The Influence of the degree of forest management on methylmercury and the composition of microbial communities in the sediments of boreal drainage ditches. Microorganisms 2022, 10(10), 1981. http://doi.org/10.3390/microorganisms10101981.

62. Thakur,; Alaba, M.O.; Rauniyar, S.; Singh, R.N.; Saxena, P.; Bomgni, A.; Gnimpieba, E.Z.; Lushbough, C.; Goh, K.M.; Sani, R.K. Text-mining to identify gene sets involved in biocorrosion by sulfate-reducing bacteria: A Semi-Automated Workflow. Microorganisms 2023, 11(1), 119. https://doi.org/10.3390/microorganisms11010119

References (References 11-12 and 30, Lines 470-475, 520-521):

11. van den Brand, T.P.H.; Roest, K.; Chen, G.H.; Brdjanovic, D.; van Loosdrecht, M.C.M. Occurrence and activity of sulphate reducing bacteria in aerobic activated sludge systems. World J Microbiol Biotechnol 2015, 31, 507–516. https://doi.org/10.1007/s11274-015-1807-4

12. Mohd ali, M.K.F.; Ismail, M.; Bakar, A.A.; Noor, N.M.; Yahaya, N.; Zardasti, L.; Sam, A.R.M. Influence of environmental parameters on microbiologically influenced corrosion subject to different bacteria strains. Sains Malaysiana 2020, 49 (3), 671-682. http://doi.org/17576/jsm-2020-4903-22

30. Li, X.H. Screening of anaerobic and facultative anaerobic sulfate-reducing bacteria in corrosive steel layer and comparison of their corrosion properties. PhD Thesis, Ocean University of China, China. 2019; pp. 1-113. (in Chinese)

Q6: English need reviison addressing the typos and mistakes

A6: We greatly appreciate your suggestions.

        The writing of the revised manuscript has been improved by MDPI language editing company and we promise that our manuscript will meet the requirements of the magazine. The certificate for English editing is provided.

Reviewer 2 Report

In this manuscript, culturable SRB was isolated and identified from the inner rust layer of three kinds of steel and from sediments. Facultative anaerobic bacteria were screened from the obtained strains, and analyses of relevant enzymes in the sulfate reduction pathway of SRB type strains were carried out. Overall, the topic is interesting. However, the method is vague and unclear, which makes it difficult for me to fully judge the preciseness of the manuscript. I have some comments to improve the quality for authors. 

1. The overall English writing needs to be carefully revised to meet the standards required by the journal. In the current manuscript, many inappropriate expressions and unprofessional words have reduced the quality of the manuscript. For instance, the main being (line 21), … included two results (line 27), et al…

2. Please explain the meaning of this sentence in detail to make it clear. “while those of facul-tative anaerobic bacteria capable of producing hydrogen sulfide included two results: only related enzymes from the dissimilatory pathway or enzymes from both pathways. Line 26-28

3. The title needs to be revised to highlight the topic more clearly.

4. The method for screening facultative anaerobic SRB need to be more specific. Line 135. From the currently described method, I cannot see the difference between this facultative anaerobic SRB and anaerobic SRB bacteria above.

5. 16SrDNA in line 127, 16SrRNA in line 143, need to be consistent throughout the manuscript.

6. line 154-155, The amino acid sequences were compared offline and online, and the amino acid sequences were edited by EditSeq software. The method is very vague. 

7. What do the figures in Tables 2 and 3 represent? Quantity or proportion? And all the legends of Tables and Figures need to be more detailed to provide more information and make self-explanation.

I have no further comment. 

Author Response

    We gratefully thanks for the precious time you spent making constructive comments. We really appreciate all your generous suggestions which have enabled us to improve our work. We have made revisions accordingly. In the following, we give a point-by-point reply to your comments and revisions were marked up using red color in the revised manuscript.

Q1: The overall English writing needs to be carefully revised to meet the standards required by the journal. In the current manuscript, many inappropriate expressions and unprofessional words have reduced the quality of the manuscript. For instance, the main being (line 21), … included two results (line 27), et al….

A1: Thank you very much for your insightful suggestions.

        The writing of the revised manuscript has been improved by MDPI language editing company and we promise that our manuscript will meet the requirements of the magazine. The revised parts in the text are shown in red. The certificate for English editing is provided.

        For instance, “the main being” was changed by “with the majority being” (Line 20); “… included two results” was changed by “included two possible ways”(line 27)

Q2: Please explain the meaning of this sentence in detail to make it clear. “while those of facultative anaerobic bacteria capable of producing hydrogen sulfide included two results: only related enzymes from the dissimilatory pathway or enzymes from both pathways. Line 26-28.

A2: We greatly appreciate your question. We apologize that our text may have caused confusion.

    The sentence has been revised to “while those of facultative anaerobic bacteria capable of producing hydrogen sulfide included two possible ways: containing the related enzymes from the dissimilatory pathway only, or containing enzymes from both dissimilatory and assimilation pathways.” in the text. (Lines 26-28).

Q3: The title needs to be revised to highlight the topic more clearly.

A3: Thank you for your rigorous review and highly valuable suggestions.

    The title of the paper has been revised to “The isolation of anaerobic and facultative anaerobic sulfate-reducing bacteria (SRB) and a comparison of related enzymes in their sulfate reduction pathways”.

Q4: The method for screening facultative anaerobic SRB need to be more specific. Line 135. From the currently described method, I cannot see the difference between this facultative anaerobic SRB and anaerobic SRB bacteria above.

A4: Thank you for pointing out the issue. We are extremely sorry for the ambiguity rising from the section.

       The difference between the two approaches for isolation methods was anaerobic or aerobic environments. The relevant parts have been rewritten in the revised manuscript and were underlined in red (Lines130-141 and Lines152-157, respectively).

Separation of anaerobic SRB:

         In a portable glove box filled with N2 (Captairpyr, Erlab company, France), which ensured that no oxygen was present in the operating environment, the prepared bacterial suspension was spread on Postgate's B (PGB) modified medium (anaerobic medium for SRB) plates [30,34]. These plates were sealed and put into 2.5 L round-bottomed vertical anaerobic culture bags with AnaeroPack (Mitsubishi Gas Chemical Co., Inc., Tokyo, Japan), then stored at 25°C for 72 h. When the medium turned black, this indicated the presence of SRB. Then, plates of suitable dilution were selected and the colonies grown on them were isolated and purified by streaking several times to obtain single colonies that could darken the medium [35]. Those strains that were able to blacken the medium and had a rotten egg smell were considered to be SRB [36], and were cultured in anaerobic Postgate's B (PGB) modified liquid medium in penicillin bottles [30] to carry out the subsequent research. (Lines 130-141, Materials and Methods 2.3)

Separation of Facultative anaerobic SRB:

        The bacteria were inoculated on modified PGB medium plates in a sterile environment, and cultured at 25°C in a biochemical incubator (SPX-150, Wanfeng Instrument Co.Ltd, China) for 72 h. Due to its cultivation under aerobic conditions, SRB strains that could grow on the plate under these conditions can be considered facultative anaerobic SRB, while bacteria that could not grow on the plate medium are considered anaerobic SRB. (Lines 152-157, Materials and Methods 2.5)

Reference (Lines 520-521, 529-534):

30. Li, X.H. Screening of anaerobic and facultative anaerobic sulfate-reducing bacteria in corrosive steel layer and comparison of their corrosion properties. PhD Thesis, Ocean University of China, China. 2019. (in Chinese)

34. Postgate, J.R. The Sulphate Reducing Bacteria. 2nd edition.; Cambridge University Press, New York, 1983; pp. 224

35. Li, X.H.; Xiao, H.; Zhang, W.; Li, Y.; Tang, X.; Duan, J.; Yang, Z.; Wang, J.; Guan, F.; Ding, G. Analysis of cultivable aerobic bacterial community composition and screening for facultative sulfate-reducing bacteria in marine corrosive steel. J Oceanol Limnol 2019, 37(2), 600-614. https://doi.org/10.1007/s00343-019-7400-1

36. Pan, J.; Shao, Z.; Cao, H.; Sheng, Y. Review on the isolation and purfication methods of sulfate reducing bacteria. J Microbiol 2007, 27(5), 79-83. https://doi.org/10.3969/j.issn.1005-7021.2007.05.018. (in Chinese)

Q5: 16SrDNA in line 127, 16SrRNA in line 143, need to be consistent throughout the manuscript.  

A5: Thank you very much for your reminding. We feel sorry for the inconvenience brought to the reviewer.

        We have checked the full text carefully and revised it in the newly submitted version (Lines 145, 162).

Q6: line 154-155, The amino acid sequences were compared offline and online, and the amino acid sequences were edited by EditSeq software. The method is very vague.

A6: We greatly appreciate your suggestions. We feel sorry for the inconvenience brought to the reviewer.

        We have already rewritten this part according to the reviewer's comments as follows:

        The genomes and proteomes of the 23 strain types were downloaded from the NCBI website and used to perform the isozyme clustering of sulfate-reduction-related genes. The amino acid sequences encoding the target enzyme were searched in the proteome of the downloaded strains and the target enzymes. BioEdit Sequence Alignment Editor software and the NCBI website (https://www.ncbi.nlm.nih.gov/) were used to perform alignment of amino acid sequences offline and online, respectively, and the amino acid sequences were edited using EditSeq software. The amino acid sequences of target enzymes encoded by different bacteria were compared using CLUSTALX 1.83 software and MEGA 6.06 software, respectively. The methods mentioned above were similar to those referenced in a previous study [30] (Lines 167-177)

Reference (Lines 520-521):

30. Li, X.H. Screening of anaerobic and facultative anaerobic sulfate-reducing bacteria in corrosive steel layer and comparison of their corrosion properties. PhD Thesis, Ocean University of China, China. 2019; pp. 1-113. (in Chinese)

Q7: What do the figures in Tables 2 and 3 represent? Quantity or proportion? And all the legends of Tables and Figures need to be more detailed to provide more information and make self-explanation.

A7: Thank you very much for your valuable question and suggestion.

        Numbers represent the numbers of isolates.

        It’s Quantity.

        All the legends of Tables and Figures have been more detailed to provide more information and make self-explanation as follows:

Table 1. Information on SRB maintained in our laboratory and SRB reviewed in the literature for the comparison of related enzymes in the sulfate reduction pathway. (Lines 181-182)

Table 2. Distribution of the numbers of culturable SRB strains in the inner rust layer of different steel materials and sediments. (Lines 196-197)

Table 3. Distribution of the numbers of isolated bacterial strains with different respiration patterns in different materials. (Lines 223-224)

Figure 1. Schematic diagram of (a) the pathway of assimilatory sulfate reduction and (b) the pathway of dissimilatory sulfate reduction. The process shared by these two pathways is sulfate ion activation by ATP sulfurase to APS, in which there are different enzymes in the assimilation pathway and the dissimilation pathway, which can be reflected in the different key enzymes involved in these two pathways. (Lines 88-92)

Figure 2. The corrosion morphology of three alloys immersed in the coastal seawaters of Sanya: (a) carbon steel (Q235) (10~15 mm thick); (b) alloy steel (Q345) (8~10 mm thick); (c) low-carbon quenched and tempered steel (921A) (10~15 mm thick). (Lines 118-120)

Figure 3. The growth of 7 SRB strains inoculated on modified PGB medium plates cultured in anaerobic and aerobic environments: (a) anaerobic environments; (b) aerobic environments. (Lines 226-227)

Figure 4. The phylogenetic tree of 7 isolated SRB strains (3 anaerobic SRB strains and 4 facultative anaerobic SRB strains) based on 16S rRNA sequences using neighbor-joining methods. Associated taxa were clustered in the bootstrap test (1000 replicates), and the bootstrap values were greater than 50%. (Lines 229-232)

Figure 5. Distribution of genes encoding different sulfate reduction pathways in bacterial strains. “+” indicates presence; “-” indicates absence. (Lines 249-250)

Reviewer 3 Report

Your paper was one of the few that I read with joy in the field of SRB  research. I am quite glad you have found several innovative findings in you research. I would like to  suggest that you could have added a separate section or something like this at the end of  your paper (perhaps under Conclusions) to list the questions that in your opinion there is an immediate need for research. 

Also, the figure that illustrates the difference between assimilatory sulfate reduction  and dissimilatory pathways is too confusing. I would suggest  to make it more understandable and also a caption more descriptive.

Author Response

    We sincerely thank you for reading our paper carefully and giving positive comments. Those comments are all valuable and very helpful for revising and improving our paper, as well as the important guiding significance to our researches. We have studied comments carefully and have made correction accordingly. In the following, we give a point-by-point reply to your comments and revisions were marked up using green- color in the revised manuscript.

Q1: I would like to suggest that you could have added a separate section or something like this at the end of your paper (perhaps under Conclusions) to list the questions that in your opinion there is an immediate need for research.

A1: Thank you very much for your insightful instructions.

       This part has been added under Conclusions (Lines 425-431) as follows:

        Further research is needed to determine the production of Dsr in anaerobic and facultative anaerobic SRB under anaerobic conditions to determine whether the isolated SRB possess the dissimilation sulfate reduction pathway necessary to produce hydrogen sulfide. Moreover, differences in the mechanisms of sulfate reduction between facultative anaerobic SRB with only the dissimilation pathway and those with both pathways, as well as functional differences and interactions between anaerobic bacteria and facultative anaerobic bacteria in metal corrosion, need to be clarified.

Q2: Also, the figure that illustrates the difference between assimilatory sulfate reduction and dissimilatory pathways is too confusing. I would suggest to make it more understandable and also a caption more descriptive.

A2: We greatly appreciate your suggestions. We apologize that our text may have caused confusion.

       We have modified the figure 1 and Figure1 legends in the text (Lines 87-92, Fig. 1).

Figure 1. Schematic diagram of (a) the pathway of assimilatory sulfate reduction and (b) the pathway of dissimilatory sulfate reduction. The process shared by these two pathways is sulfate ion activation by ATP sulfurase to APS, in which there are different enzymes in the assimilation pathway and the dissimilation pathway, which can be reflected in the different key enzymes involved in these two pathways.

Round 2

Reviewer 2 Report

The author used 'Postgate's B (PGB) modified medium' in the manuscript, please clarify the formula, as it is a modified formula.

Author Response

Q1: The author used 'Postgate's B (PGB) modified medium' in the manuscript, please clarify the formula, as it is a modified formula.

A1: Thank you very much for your insightful instructions.

       We have supplemented this part in the revised manuscript accordingly (Lines 133-136, in red color) as follows:

        Postgate's B (PGB) modified medium (anaerobic medium for SRB) plates Postgate's B (PGB) modified medium (anaerobic medium for SRB) plates [30,34] containing KH2PO4 (0.5 g·L-1), NH4Cl (1.0 g·L-1), Na2SO4 (1.0 g·L-1), CaCl2·2H2O (0.1 g·L-1), MgSO4·7H2O (2.0 g·L-1), C3H5O3Na (80%) (3.5 mL·L-1), yeast extract (1.0 g·L-1), FeSO4·7H2O (0.5 g·L-1), agar (20.0 g·L-1), Vitamin C (0.1 g ·L-1) and H-Cys-OH·HCl (0.5 g ·L-1) with a pH between 7.0 and 7.2.

Reference (Lines 523-524, 532):

30. Li, X.H. Screening of anaerobic and facultative anaerobic sulfate-reducing bacteria in corrosive steel layer and comparison of their corrosion properties. PhD Thesis, Ocean University of China, China. 2019. (in Chinese)

34. Postgate, J.R. The Sulphate Reducing Bacteria. 2nd edition.; Cambridge University Press, New York, 1983; pp. 224